# The Drosophila amyloid precursor protein homologue mediates neuronal survival and neuroglial interactions

**Irini A. Kessissoglou**[1], **Dominique Langui**[1], **Amr Hasan**[2], **Maral Maral**[1], **Suchetana B. Dutta**[1,2], **Peter Robin Hiesinger**[2], **Bassem A. Hassan**[1,2]*

**1** Paris Brain Institute, Hôpital Pitiê-Salpêtrière, Inserm U 1127, CNRS UMR, Sorbonne Université, Paris, France, **2** Division of Neurobiology, Institute for Biology, Freie Universität Berlin, Berlin, Germany

* bassem.hassan@icm-institute.org

**Data Availability Statement:** All relevant data are within the paper and its Supporting Information files

## Abstract

The amyloid precursor protein (APP) is a structurally and functionally conserved transmembrane protein whose physiological role in adult brain function and health is still unclear. Because mutations in APP cause familial Alzheimer's disease (fAD), most research focuses on this aspect of APP biology. We investigated the physiological function of APP in the adult brain using the fruit fly *Drosophila melanogaster*, which harbors a single APP homologue called APP Like (APPL). Previous studies have provided evidence for the implication of APPL in neuronal wiring and axonal growth through the Wnt signaling pathway during development. However, like APP, APPL continues to be expressed in all neurons of the adult brain where its functions and their molecular and cellular underpinnings are unknown. We report that APPL loss of function (LOF) results in the dysregulation of endolysosomal function in neurons, with a notable enlargement of early endosomal compartments followed by neuronal cell death and the accumulation of dead neurons in the brain during a critical period at a young age. These defects can be rescued by reduction in the levels of the early endosomal regulator Rab5, indicating a causal role of endosomal function for cell death. Finally, we show that the secreted extracellular domain of APPL interacts with glia and regulates the size of their endosomes, the expression of the Draper engulfment receptor, and the clearance of neuronal debris in an axotomy model. We propose that APP proteins represent a novel family of neuroglial signaling factors required for adult brain homeostasis.

## Introduction

Early-onset familial Alzheimer's disease (fAD) is caused by several mutations either in the amyloid precursor protein (APP) or in the Presenilin (PSEN-1 and PSEN-2) genes [1,2]. APP is a functionally and structurally conserved transmembrane protein, present in both invertebrates like *Caenorhabditis elegans* and *Drosophila melanogaster* [3,4] and mammals [5–7]. APP undergoes 2 competing proteolytic processes: the amyloidogenic processing where it is internalized into endosomes and cleaved by β-secretase and subsequently γ-secretase releasing sAPPβ, the amyloid-β (Aβ) oligomers and APP intracellular domain (AICD), and the non-

**Funding:** This work was supported by the program "Investissements d'avenir" ANR-10-IAIHU-06 of the Agence Nationale de la Recherche (https://anr.fr/) (to BAH), The Einstein-BIH Visiitng Fellow program EVF-BIH-2015-236-2 (https://www.einsteinfoundation.de/en/programmes/einstein-bih-visiting-fellow) (to BAH), the Paul G. Allen Frontiers Group Allen Disntinguished Investigator Award 12202 (https://alleninstitute.org/what-we-do/frontiers-group) (to BAH), and the Roger De Spoelberch Foundation 2019 Prize (https://www.fondation-roger-de-spoelberch.ch/en) (to BAH), the National Institutes of Health (https://www.nih.gov) RO1EY018884 (to PRH) and the German Research Foundation (https://www.dfg.de) (SFB 958, SFB186) (to PRH)and Freie Universität Berlin (to PRH). The funders had no role in study design, data collection and analysis, decision to publish, or preparation of the manuscript. PRH received salary form FU Berlin.

**Competing interests:** The authors have declared that no competing interests exist.

**Abbreviations:** Aβ, amyloid-β; AD, Alzheimer's disease; ADAM, A Disintegrin and Metalloprotease; AICD, APP intracellular domain; APPβCTF, APP β C-terminal fragment; APP, amyloid precursor protein; APPL, APP Like; BSA, bovine serum albumin; Dcp-1, Cleaved Drosophila Death caspase protein-1; DF, double fluorescent; dT-APPL, double-tagged form of APPL; fAD, familial Alzheimer's disease; flAPP, full-length APP; flAPPL, full-length APPL; GFP, green fluorescent protein; iPSCs, induced pluripotent stem cells; JNK, c-Jun N-terminal kinase; JIP, JNK interacting proteins; LOF, loss of function; Myr-DF, myristoylated double fluorescent; ORN, olfactory receptor neuron; PAT-1, Protein interacting with APP tail 1; PBS, phosphate buffered saline; PCP, planar cell polarity; PSEN, Presenilin; RNAi, RNA interference; RT, room temperature; SAPPL, secreted portion of APPL; TEM, transmission electron microscopy; Vang, VanGogh.

amyloidogenic processing where APP is cleaved on the cellular membrane by α-secretase and subsequently γ-secretase releasing sAPPα, the P3 domain and AICD [8].

fAD mutations result in the enhancement of the amyloidogenic processing of APP and hence in not only an increased release of Aβ oligomers, but also a reduced production of sAPPα [9] and potentially other unknown effects on APP's physiological function, such as the balance between its intracellular and extracellular activities. The accumulation of Aβ oligomer aggregates is also present in the brain of patients with sporadic Alzheimer's disease (AD), forming the Aβ plaques and leading to the hypothesis that Aβ plaques are the main cause of the disease [10]. However, thus far all anti-amyloid treatment, although often successful in reducing the amyloid load, have failed to improve AD symptoms [11]. This raises the need for a better understanding of the physiological function of APP in order to design better future treatment.

In vitro loss of function (LOF) studies on human or mouse APP revealed its involvement in a variety of functions related to neuron biology, such as neural stem cell proliferation, differentiation, and neurite outgrowth of hippocampal neurons [12]. Moreover, it seems to have a role in synapse formation, as a synaptic adhesion molecule [13]. APP's conserved intracellular domain interacts with many protein-signaling pathways such as the c-Jun N-terminal kinase (JNK) to induce cell death [14], X11/JNK interacting proteins (JIP) to activate cell differentiation [15], and with Fe65 to modulate gene transcription [16].

In *D. melanogaster*, APP Like (APPL) is the single homologue of the human neuronal APP$_{695}$ sharing 30% homology at the amino acid level [17]. In vivo LOF studies have demonstrated that APPL is involved in axonal outgrowth during development [18], axonal transport of vesicles or mitochondria [19,20], synapse formation at the neuromuscular junction [21], and long-term and working memory formation [22,23]. Moreover, it has been shown that the secreted portion of APPL (SAPPL) has a neuroprotective function by rescuing vacuole formation in the brain of neurodegenerative mutant flies through its binding to the full-length APPL (flAPPL) [24]. Finally, like in mammals, APPL acts as a receptor and interacts with G$_0$ proteins, cell adhesion molecules, and intracellular modulators like the dX11/Mint protein, Tip60, and Fe65 [25–27].

An interesting observation is that most APP LOF studies in a plethora of neuronal processes and molecular mechanisms reveal relatively mild phenotypes with relatively low penetrance. Combined with the fact that neuronal forms of APP are expressed throughout the brain, this suggests that APP is a homeostasis factor required for the brain to develop correctly, remain stable, and counteract internal and external perturbations. The nervous system encounters several types of genetic mutations and environmental perturbations that can cause organelle stress and cell death and finally can lead to developmental, age, or stress-associated disorders. To counteract this, animals have evolved a defense homeostatic signaling system, composed of protein chaperones and transcriptional mechanisms [28] involving both neurons and glial cells such as astrocytes, Schwann cells, and oligodendrocytes [29]. However, the molecules that neurons use to communicate homeostatic signals to glia remain largely unknown.

A major homeostatic cellular mechanism is the endolysosomal recycling and degradation pathway [30]. This pathway ensures that cellular cargo is properly recycled between the membrane and various organelles or degraded to maintain protein homeostasis and cellular health. A study on primary neurons revealed that an APP intracellular binding protein, PAT-1, regulates the number of early endosomes and endocytosis [31]. Recently, 2 studies revealed that induced pluripotent stem cell (iPSC)-derived human neurons with either APP or PSEN1 fAD knock-in mutations show enlarged and defective early endosomes and lysosomes [32,33]. Therefore, this might suggest a role for APP in the neuronal endolysosomal pathway.

To investigate the in vivo role of APP in neuronal homeostasis, we used *Drosophila* as a model organism and investigated the consequences of the deletion of its homologue, the *Appl*

gene. We report that loss of APPL results in the increased accumulation of apoptotic cells in the brain at a critical young age. We link this accumulation to defects in the endolysosomal pathway in both neurons and glia and show that APPL is required for neuroglial communication.

## Results

### APPL is required for neuronal survival in young adult flies

To investigate the implication of APPL in brain health of adult flies, we started with quantifying the survival of APPL null flies (*appl^d*) [34] compared to genetic background controls (Canton S) at different stages of their life span. As previously reported [24], *appl^d* flies die significantly earlier than their control counterparts in a sex-independent manner starting at 2 to 3 weeks of age (Fig 1A). This suggests that loss of APPL compromises survival at an early age. Because *Drosophila* APPL is an exclusively neuronal protein [17], we asked whether neuronal health is compromised in APPL mutants during the first 3 weeks of life. We measured the cell death load in the brain of *appl^d* and controls at 2, 7, 21, and 45 days of age. To quantify the number of dying cells at any given moment, we stained whole mount brains with Cleaved Drosophila Death caspase protein-1 (Dcp-1), the homologue of human Caspase 3, and manually quantified the Dcp-1 positive cells across the entire central brain (Fig 1B and 1B'). In both genotypes, 2-day-old flies show significant cell death in their brain due to ongoing brain remodeling [35]. By 7 days of age, however, there is a sharp drop in the number of apoptotic cells in controls. In contrast, the drop in apoptosis is significantly reduced in *appl^d* flies, with an average of 7 to 8 apoptotic cells per brain at any given time point. Counter staining with the neuronal marker Elav and the glial marker Repo showed that all dying cells detected were neurons (Fig 1B''–1B''''). With age, at 21 and 45 days old, both control and *appl^d* flies show a similar increase in apoptotic cells (Fig 1C). These data suggest that loss of APPL renders neurons particularly sensitive during the first week of life. APPL is only detectable in neurons, although some reports have claimed it may be expressed in glia [36]. To test whether APPL expression in neurons is required for their survival at 7 days old, we knocked down the expression of APPL using the UAS-Gal4 system expressing APPL RNA interference (RNAi) only in neurons using the pan-neuronal *nSyb-Gal4* driver. This resulted in significantly more apoptotic neurons in 7-day-old *Appl* knock-down flies compared to controls, similar to *appl^d* flies (Figs 1D and S1A–S1F'). Furthermore, re-expressing flAPPL or only SAPPL specifically in neurons in an *appl^d* background using the pan-neuronal *nSyb-Gal4* driver, significantly rescued the increased number of apoptotic cells in the brain of *appl^d* flies (Fig 1E). In contrast, overexpressing SAPPL in a control background caused the presence of significantly more apoptotic cells in the brain of 7-day-old flies (Fig 1F).

In summary, our data show that loss of APPL in neurons results in excessive neuronal death during the first week of life and a corresponding reduction in life starting 1 to 2 weeks later. These results also highlight the importance of maintaining the expression of APPL at physiological levels. We next asked by what mechanism APPL acts to protect neurons and flies from premature death.

### APPL regulates the size and number of neuronal early endosomes

We have previously shown that APPL is a neuronal modulator of the Wnt planar cell polarity (PCP) pathway for the axonal outgrowth during development. Specifically, loss of APPL sensitizes growing axons to reduction in Wnt PCP signaling and renders the PCP core protein, VanGogh (Vang), haploinsufficient [18]. We started by examining the genetic interaction between *appl* and *vang* by removing 1 copy of *vang* in the *appl^d* background and measuring

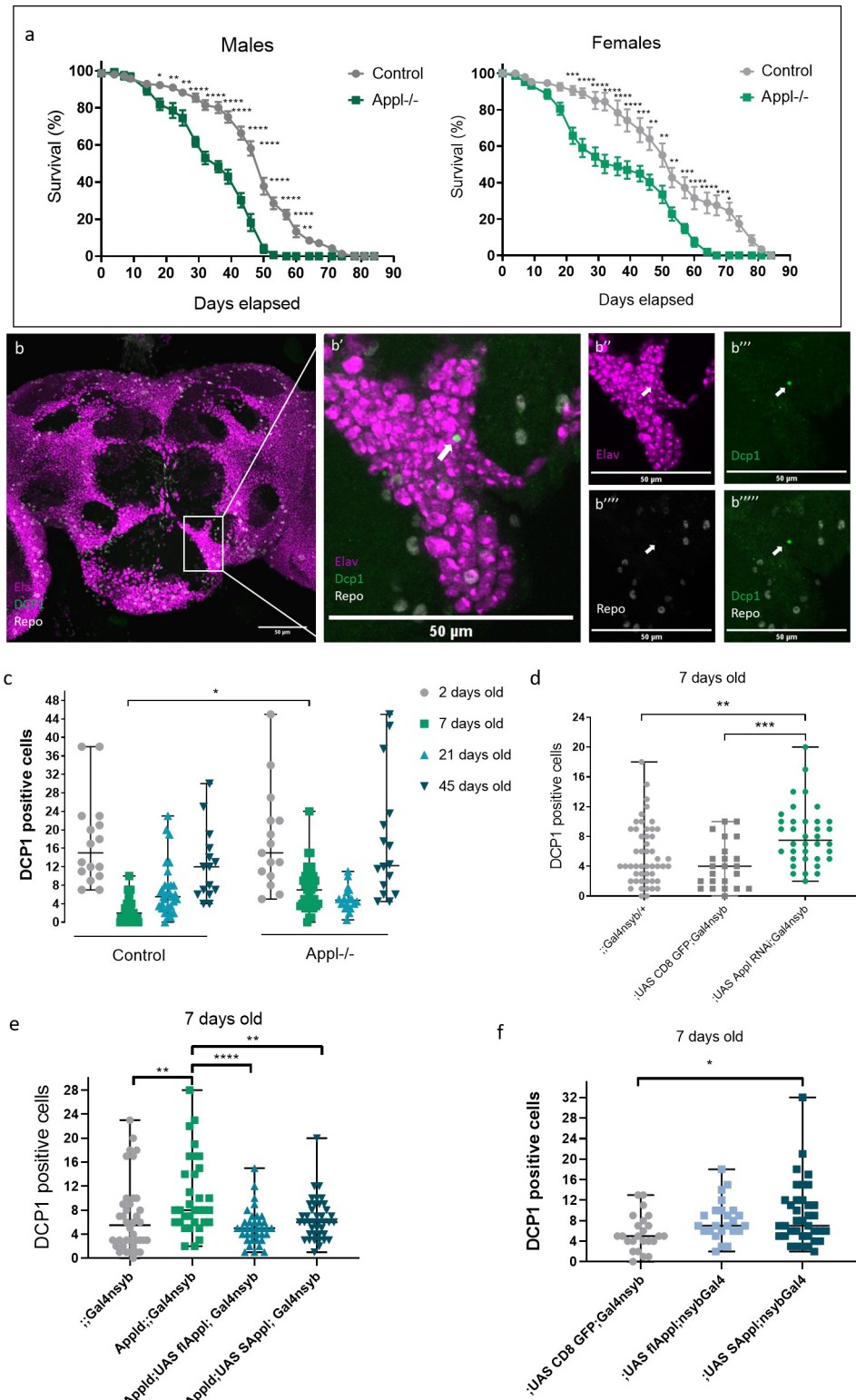

**Fig 1. Loss of APPL increases early age mortality rate and apoptotic neuronal death.** (a) These survival curves represent the life span of appld (APPL-/-) flies compared to control, Canton S flies. $n$ = 10 groups of 15 flies each for every condition (CS males, CS females, APPL-/- males and APPL-/- females). Total $n$ = 150 flies per condition. Two-way ANOVA with Sidak multiple comparison test $F_{(24, 450)}$ = 4.25, df = 24, $^*p < 0.05$, $^{**}p < 0.01$, $^{***}p < 0.0001$. (b) Confocal sections of an appld/Y;; brain stained with anti-Elav (magenta) to mark the neurons, anti-Cleaved Drosophila

Dcp-1 (green) to mark the apoptotic cells, and anti-Repo (white) to mark the glial cells. In the higher magnification pictures on the right panels, we can notice that the Dcp-1 marked cell (white arrow) colocalizes with Elav (b" and b'") but not with Repo (b""and b""'). (c) This graph shows the number of apoptotic cells in the central brain of Control and APPL-/- flies at different ages: 2, 7, 21, and 45 days old. Each data point represents the number of apoptotic cells, Dcp-1 positive cells, in a single brain. For the analysis of this data, we used 2-way ANOVA with Bonferroni multiple comparison test $F_{(3,189)} = 3.086$, df = 3, *$p = 0.027$. (d) Focusing on the 7-day-old time point, which showed significant difference in the previous graph (c), we now knock down the expression of APPL only in neurons using the *yw;UAS APPL RNAi (y+); Gal4nsyb* and find a similar increase in the number of apoptotic cells comparing to the controls: *yw;;Gal4nsyb/+* and *w-;UAS CD8 GFP;Gal4nsyb*. One-way ANOVA: $F_{(2,106)} = 0.51$, df = 2, **$p = 0.0078$, ***$p = 0.0005$. (e) Rescue experiment. Reexpression of the flAPPL or only the SAPPL in an APPL null background and quantifying the number of apoptotic cells remaining in the brain of the flies at 7 days of age. One-way ANOVA: $F_{(3, 143)} = 6.712$, df = 3, **$p = 0.0036$, **$p = 0.0095$, ****$p < 0.0001$. (f) Overexpression of the flAPPL and only the SAPPL in a control background in all neurons. One-way ANOVA: $F_{(2, 82)} = 3.967$, df = 2, *$p = 0.0120$. Underlying data can be found in the S1 Data file. APPL, APP Like; CS, Canton S; Dcp-1, Death caspase protein-1; flAPPL, full-length APPL; SAPPL, secreted portion of APPL.

neuronal death at 7 days of age. In contrast to the developmental effect on axon growth, we found no effect on the number of apoptotic neurons (S2 Fig) suggesting a different mechanism.

Studies on fAD mammalian models have described an increased activation of autophagy prior to the appearance of Aβ aggregates [37]. Our in vivo data coincide with the fAD phenotypes as we also observed an increased amount of Atg8 puncta in the central brain of 7-day-old *appl$^d$* flies (S4A and S4B Fig). We conclude that LOF of APPL also causes an accumulation of autophagosomes. However, alterations in autophagy were not linked with the early onset increase in apoptotic cell death, as an Atg7 mutation in *appl$^d$* background flies did not impact the 7-day-old cell death phenotype seen in *appl$^d$* flies (S4C Fig).

A number of observations suggest a tight link between the APP family of proteins and endolysosomal trafficking. First, both human APP and fly APPL carry highly conserved endocytic motifs in their intracellular domains [38], which interact with proteins involved in endocytosis [31]. Second, APP has been implicated in the regulation of the endocytosis of cell surface receptors [39]. Third, as the endolysosomal pathway is involved in APP's and APPL's cleavage by β- and γ-secretases, perturbations of the endolysosomal pathway can have negative repercussions in the proteolytic processing of APP and hence the amount of Aβ produced [40]. Fourth, 2 recent in vitro studies showed evidence for the development of enlarged early endosomes and lysosomes in human iPSCs with various fAD mutations in the *APP* and *PSEN1* genes [32,33]. We therefore investigated whether loss of APPL causes defects in endolysosomal function in the brain.

We used an acidification-sensing double fluorescent (DF) probe, composed of a pH-sensitive green fluorescent protein (GFP) (pHluorin) and pH-resistant mCherry fused to a myristoylated residue to track all plasma membrane trafficking (myr-DF) [41]. This probe allows the tracking of the trafficking of membrane cargo through the endolysosomal pathway. In neutral pH vesicles, like early endosomes, the probe will fluoresce in both green and red channels, whereas in acidic vesicles, such as late endosomes and lysosomes, the GFP signal will be quenched and the probe will fluoresce only in red (Fig 2A). Differences in fluorescence values between controls and mutant would indicate potential defects in endolysosomal trafficking. The myr-DF probe was expressed in all neurons, using the *nSyb-Gal4* driver, in control and *appl$^d$* flies (Fig 2C and 2C'). We focused our imaging and quantifications on 2 easy-to-identify neuronal populations: the Kenyon cells of the mushroom body and the Projection Neurons of the antennal lobes (Fig 2B). In live confocal imaging data of 7-day-old flies, due to the diffused green signal of pHLuorin, the probe does not distinguish between early and late endosomes, but permits to measure the quantity and volume of the endolysosomal compartments. This analysis revealed the presence of significantly enlarged endolysosomal compartments in the neurons of *appl$^d$* flies compared to those of controls (Figs 2C, 2D, S3A–S3C and S3E).

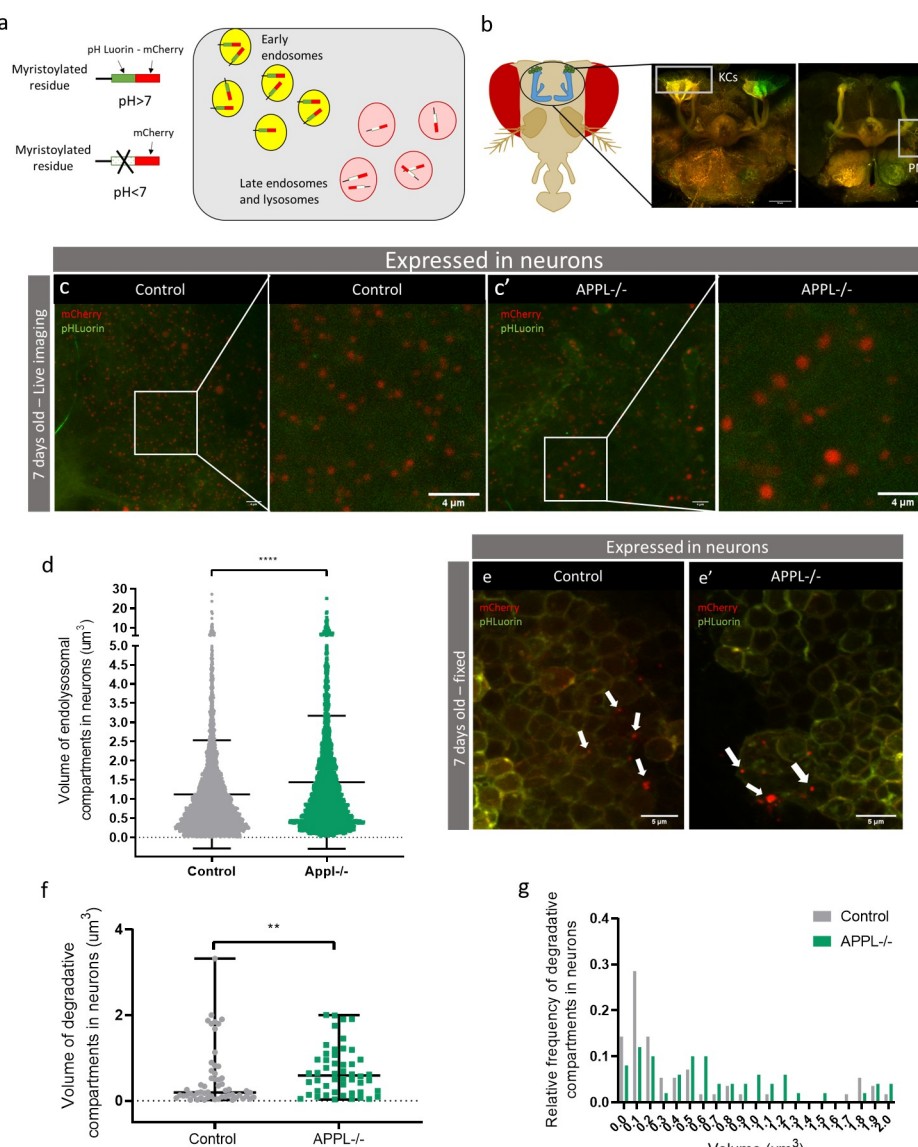

**Fig 2. Loss of APPL causes enlarged endolysosomal compartments in neurons.** (a) Schematic showing the DF probe composed of a pH-sensitive Luorin and pH-resistant mCherry. This probe gets tagged on myristoylated general plasma membrane proteins. When inside an early endosome, it emits a yellow signal and as soon as the protein cargo is inside an acidic vesicle, it produces a red signal. (b) Schematic of the head of a fly. The fluorescent images on the right highlight the areas that were imaged: the KCs and PNs. (c and c') Images from live snapshots of the KCs of adult, 7-day-old fly brains expressing the DF probe only in neurons. The left panel and its close-up show a control: *w*;UAS myr mCherry pH Luorin; nsyb Gal4* fly; and the right panel and its close-up show an *appl^d* mutant: *APPLd; UAS myr mCherry pH Luorin; nsybGal4*. (d) Quantification of the volume of endolysosomal compartments (um3). $n = 5$ brains per genotype,[****]$p < 0.0001$, Mann–Whitney post hoc test. (e and e') These confocal slices represent the same area of KCs but this time from a fixed tissue of control: *w*;UAS myr mCherry pH Luorin; nsyb Gal4* and mutant: *APPLd; UAS myr mCherry pH Luorin; nsybGal4* 7-day-old fly brains. The white arrows show the acidic degradative compartments. (f) The volume of these degradative compartments is significantly higher in *appl^d* flies, [**]$p = 0.006$, Mann–Whitney post hoc test, $n = 11$. (g) A histogram of the relative frequency of degradative compartments in neurons, $n = 11$. Underlying data can be found in the S1 Data file. APP, amyloid precursor protein; DF, double fluorescent; KCs, Kenyon cells; PNs, Projection neurons.

Live imaging data could only inform us about the trafficking of the protein cargo to an acidic vesicle, with a pH below 6, but not its degradation inside this vesicle [41]. To quantify the effect of APPL LOF on the degradation of plasma membrane protein cargo, we quantified

red-only compartments in fixed tissue, where irreversibly damaged pHLuorin leads to the selective loss of green fluorescence [41]. Results from 7-day-old fixed brains showed a marginal but not significant increase in the number of degradative compartments between control and $appl^d$ brains (Figs 2E and 2E' and S3D). However, the volume of degradative compartments in $appl^d$ flies was significantly larger (Fig 2F and 2G). Together, these analyses evoke the possible enlargement of late endosome–like vesicles, suggesting a deficit in the regulation of the volume of endolysosomal compartments in $appl^d$ flies.

To further investigate these potential defects at higher resolution, we used transmission electron microscopy (TEM) (Fig 3A and 3B). Whereas the overall size of the neuronal cell body did not differ between mutants and controls (Fig 3C), we noted the presence of enlarged

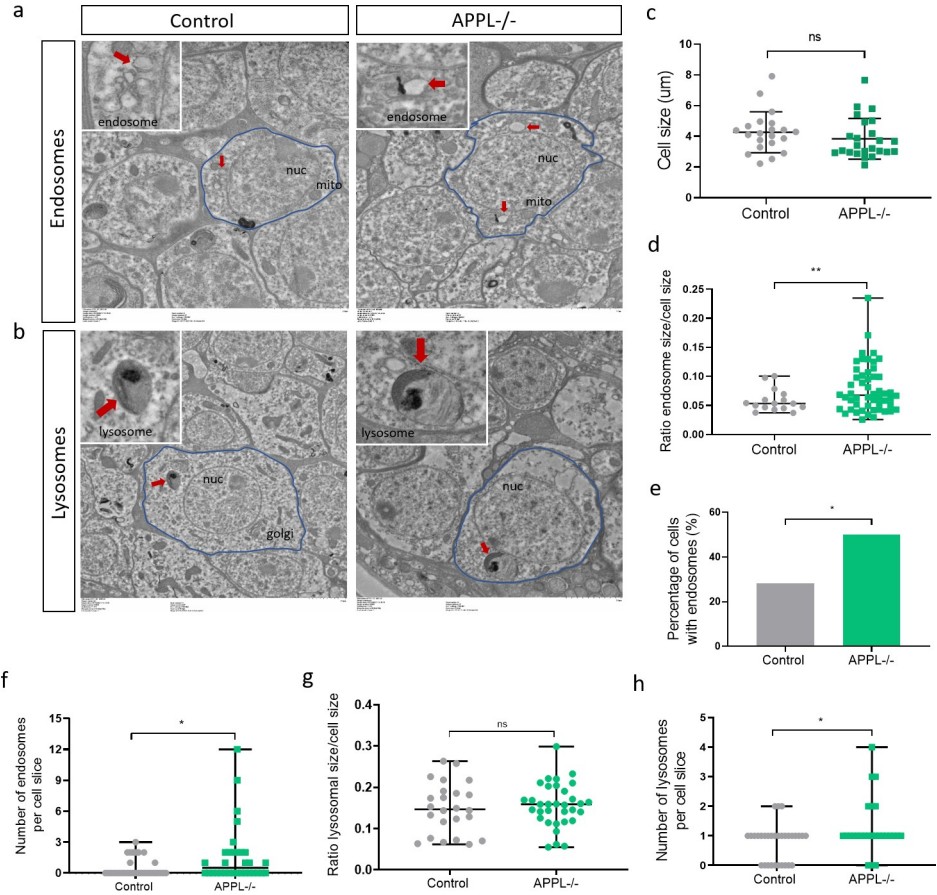

**Fig 3. APPL regulates the size of endosomes in neurons.** (a) TEM horizontal sections of the cortical region of a 7-day-old fly brain showing neuronal cell bodies (circled in blue) and their organelles. We can observe that there are more and enlarged early endosome–like vesicles (red arrow) in the brain of APPL-/- flies $w^*appl^d/Y;;$ comparing to control +/+;+/+;+/+. $n = 3$ brains per genotype and a total of 32 cells analyzed per genotype. (b) The size of lysosomes (red arrow) seem to not be affected in APPL-/- flies. (c) This graph shows that the cell size is the same between control and APPL-/- flies at 7 days old. $n = 3$ brains per genotype and a total of 32 cells analyzed per genotype. (d–f) These graphs present the difference in size between the early endosome–like vesicles seen in APPL-/- and control fly brains and the increased prevalence of endosomes in 7-day-old APPL-/- fly brains; $n = 3$ brains per genotype and a total of 32 cells analyzed per genotype. Statistical analysis was done using (d) Welch $t$ test $^{**}p = 0.0080$, (e) Binomial test: $^*p = 0.0141$, and (f) Welch $t$ test $^*p = 0.0412$. (g) This graph shows that the size of lysosomes is not affected by the absence of APPL. (h) This graph shows that there are significantly more lysosomes per cell slice in APPL-/- flies comparing to Canton S. Unpaired $t$ test $^*p = 0.018$. Underlying data can be found in the S1 Data file. APPL, APP Like; mito, mitochondria; ns, not significant; nuc, nucleus; TEM, transmission electron microscopy.

clear-single membraned endosome-like vesicles in *appl^d* neurons (Fig 3A and 3D). In addition, there were more of them per section than in controls (Figs 3E, 3F, S5A and S5B). On the other hand, lysosomal size did not seem to be affected in *appl^d* flies (Figs 3B and 3G, S5C and S5D), and there was a marginal but significant increase in lysosomal number per section (Fig 3H). To verify that APPL expression in neurons is required for the regulation of endosomal size, we knocked down the expression of APPL by expressing APPL RNAi only in neurons using the pan-neuronal *nSyb-Gal4* driver. Knocking down the expression of APPL resulted in significantly more cells containing early endosome–like vesicles and significantly larger vesicles compared to controls, similar to *appl^d* flies (S5E–S5G Fig). Moreover, using the same UAS-Gal4 system, we also re-expressed, specifically in the neurons of *appl^d* flies, the flAPPL and only the SAPPL. Interestingly, re-expressing the flAPPL in *appl^d* flies reduced the accumulation of enlarged early endosome–like vesicles to control levels, confirming that the phenotype we observed was caused by the loss of the flAPPL (Fig 4A–4C).

These data confirm the presence of defects in the neuronal endolysosomal pathway in the absence of APPL and suggest that these defects arise mostly in endosomes and are specific to the loss of the flAPPL in neurons suggesting an endogenous regulation of the size of endosomes by APPL.

Our data so far suggest a defective accumulation of enlarged endosomes in neurons of *appl^d* flies. The trafficking of cargo from the membrane to early endosomes is regulated by the Rab5 GTPase. To investigate whether defects in early endosomes cause the increase in the number of dying neurons in *appl* mutant brains, we first examined the expression of Rab5 GTPase in 7-day-old *appl^d* flies. Results revealed significantly higher levels of Rab5 in *appl* mutant brains compared to controls (Fig 5C and 5D), similar to what has previously been reported in AD patients' fibroblasts [42]. Next, we removed 1 copy of the *rab5* gene in an *appl* null background. This completely rescued the neuronal cell death phenotype at 7 days of age back to control levels (Fig 5A and 5B). We conclude that reducing the trafficking to early endosomes in an *appl* null condition re-equilibrates the system and rescues the functioning of the endolysosomal pathway.

To test whether the rescue effect is specific to the early endosomal stage, we removed a copy of the gene encoding the late endosomal marker Rab7 in *appl^d* mutant background. In contrast to reduction of Rab5 levels, this failed to rescue the number of apoptotic cells found in the brains of 7-day-old *appl^d* flies (S4D Fig), and indeed significantly worsened the life span of the flies relative to controls (S4E Fig), consistent with a role for Rab7 itself in neurodegeneration [43].

Our observations suggest that in the absence of APPL, neurons accumulate enlarged vacuole-like endosomal compartments, possibly due to the dysregulation of early endosomes, resulting in neuronal death in the young adult brain. What is intriguing, however, is why these dying neurons accumulate to a sufficient level as to be detectable instead of being cleared by glial cells. We asked whether this accumulation of dying neurons is due only to the dysregulated endolysosomal network, or also to defective glial clearance.

## The extracellular domain of APPL is secreted by neurons and interacts with glia

APPL is a transmembrane protein that is cleaved resulting in a secreted form, SAPPL. To explore the expression and secretion pattern of APPL, we generated a double-tagged form of APPL (dT-APPL) with GFP intracellularly (C-terminally) and mCherry extracellularly (N-terminally) (Fig 6A). To study the distribution and spread of SAPPL, we expressed dT-APPL strictly in the retina and imaged the entire brain at different stages of pupal development and

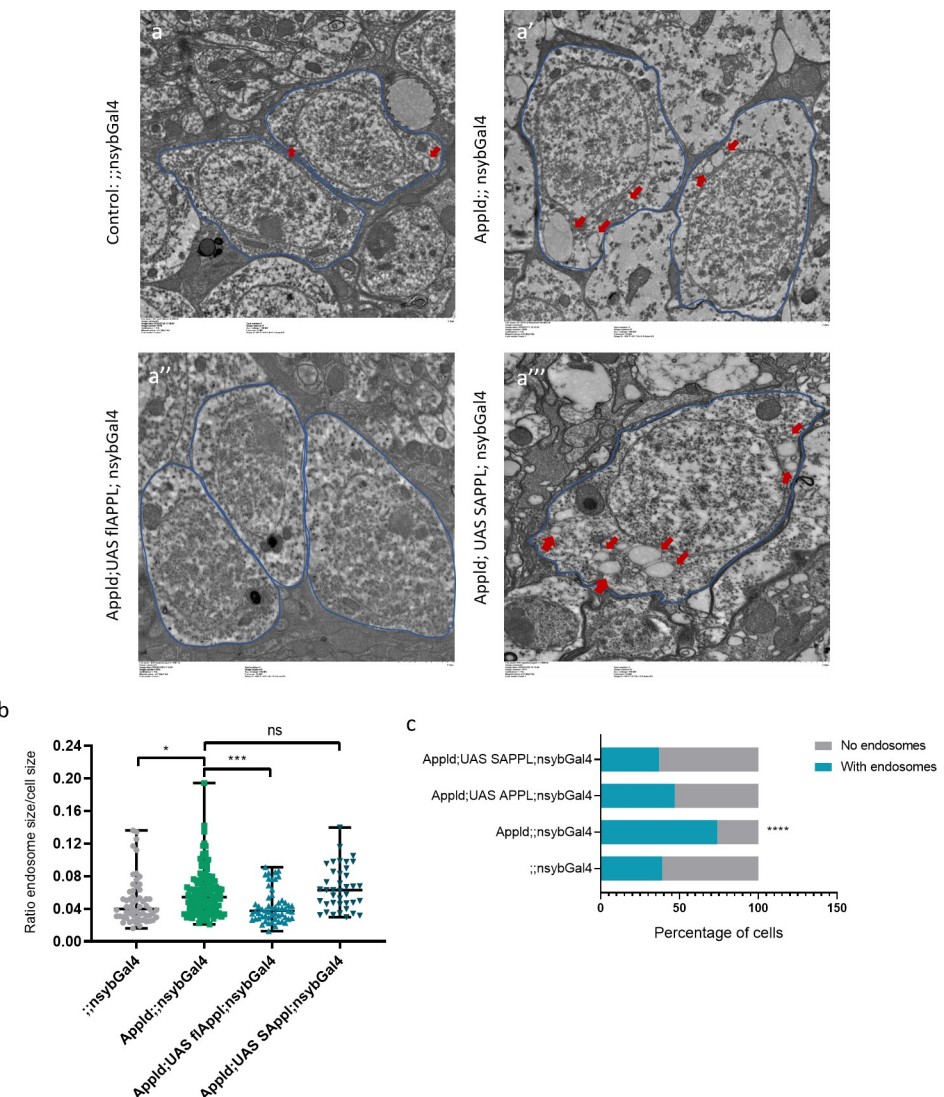

**Fig 4. Rescue of the endosome enlargement. (**a–a''') TEM horizontal sections of the cortical region of 7-day-old fly brains showing neuronal cell bodies (circled in blue) and the early endosome–like vesicles marked with a red arrow. Each panel is a different genotype: (a) control,*;; nsybGal4*, (a') Appl-/-, *Appld;;nsyb Gal4*, (a'') reexpression of the flAPPL, *Appld; UAS flAPPL; nsybGal4*, and (a''') only the SAPPL *Appld; UAS SAPPL; nsybGal4* individually in an APPL null background. *n* = 3 to 5 brains per genotype and approximately 60 cells analyzed per genotype. (b) This graph represents the quantification of the size of the early-endosome–like vesicles. Reexpressing the flAPPL rescued the increased size of early-endosome–like vesicles seen in *Appld;;nsybgal4* flies, *n* = 3 to 5 brains per genotype and approximately 60 cells analyzed per genotype $^*p$ = 0.0191, $^{***}p$ = 0.0005. (c) The significantly higher percentage of cells containing early-endosome–like vesicles was also rescued when reexpressing the flAPPL and only the SAPPL, Fisher exact test $^{****}p$ < 0.0001. Underlying data can be found in the S1 Data file. APPL, APP Like; flAPPL, full-length APPL; ns, not significant; SAPPL, secreted portion of APPL; TEM, transmission electron microscopy.

in the adult (S6A Fig). Whereas the intracellular part of the *appl* protein (GFP) remained inside photoreceptors, SAPPL (mCherry) gradually spread throughout the whole brain starting from 80H after puparium formation and remained so in adults (S6A–S6C Fig). Moreover, SAPPL was taken up by glia (S6C Fig). To ascertain that glial uptake of SAPPL was not a consequence of APPL overexpression in the presence of the endogenous protein, we repeated this experiment by expressing the dT-APPL in all post-mitotic neurons of *appl* null flies. Again,

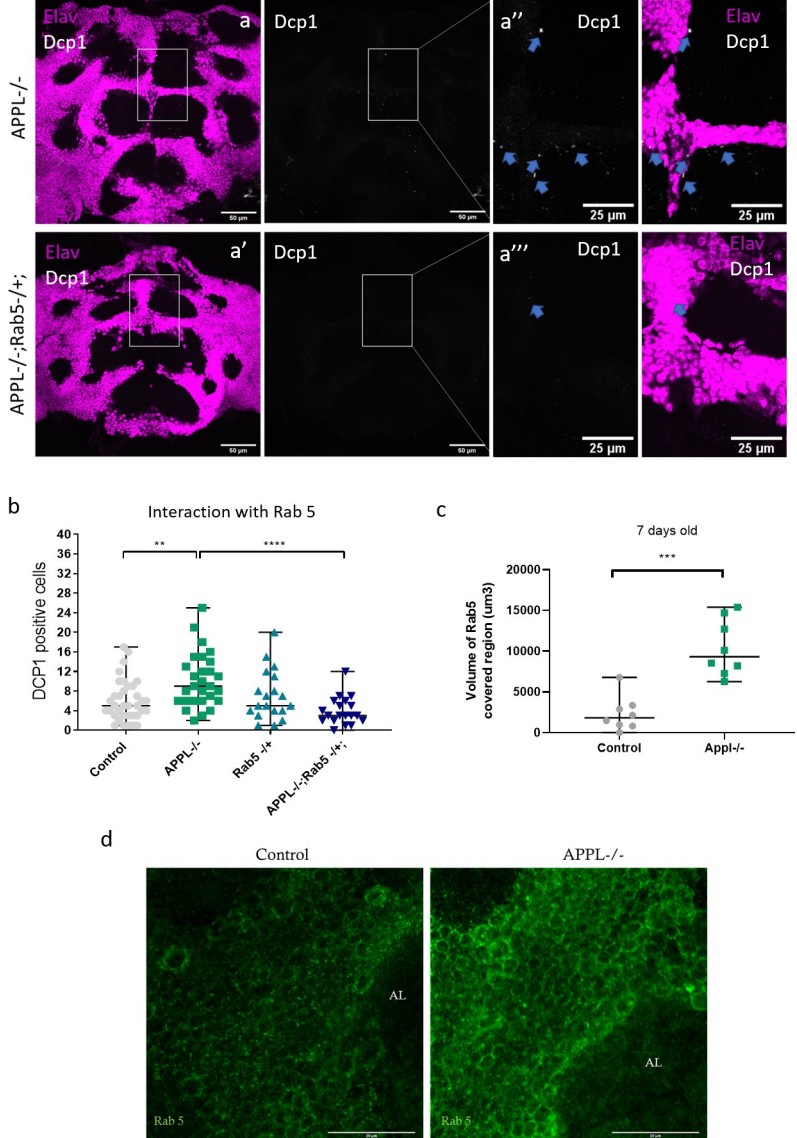

**Fig 5. Causal link between the overactivation of Rab5 and the increased neuronal cell death.** (a–a'") Confocal sections of the central brain of 7-day-old APPL-/- flies and APPL-/- flies heterozygous for Rab5, stained with the neuronal marker elav (magenta) and the apoptotic marker dcp-1 (white). Blue arrows point to the dcp-1 positive cells. (b) Quantification of apoptotic cells in the central brain of control $w^*/Y;+/+;+/+$, $w^* appl^d/Y;;$, $w^*/Y;Rab5 KO/+;$ and $w^* appl^d/Y;Rab5 KO/+;$. Reducing 1 copy of Rab5 in an APPL-/- background shows a significant reduction in the number of Dcp-1 positive cells observed in APPL-/- flies at 7 days old. One-way ANOVA: F(3,98) = 7.987, df = 3, $^{**}p = 0.0013$ $^{****}p < 0.0001$. (c) Graph representing the quantification of the volume of Rab5-covered region (um3) in the central brain of 7-day-old APPL-/- flies and control flies. There is significantly more volume of Rab5-covered region (more early endosomes or larger early endosomes) in APPL-/- flies in comparison to control. $n = 8$ per genotype, Mann–Whitney test $p^{***} = 0.0003$. (d) Confocal sections of the central brain of 7-day-old APPL-/- flies and control flies stained with anti-Rab5 (green). Underlying data can be found in the S1 Data file. AL, antennal lobes; APPL, APP Like; Dcp-1, Cleaved Drosophila Death caspase protein-1.

while the intracellular part of APPL remained in neurons, SAPPL was localized both in neurons and in glia (Fig 6B and 6B'"). To confirm the functionality of the dT-APPL, we re-expressed this form of APPL in $appl^d$ flies and found that this rescues the developmental axonal defect phenotype in the mushroom bodies reported previously (S6D and S6E Fig) [18].

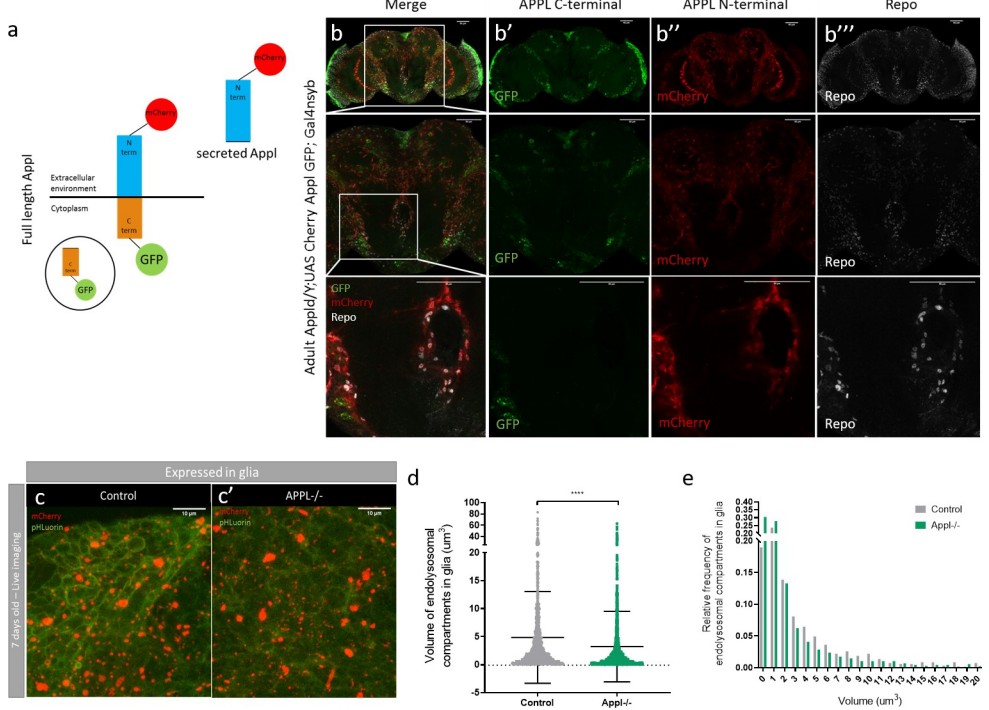

**Fig 6. SAPPL interacts with glia and affects their endolysosomal network. (a)** Schematic representation of the double-tagged APPL construct with GFP on the intracellular part and mCherry on the extracellular part. **(b–b''')** Confocal sections of an adult APPL null fly brain expressing the double-tagged construct *Appldw*/Y; UAS CherryApplGFP / +; nsyb Gal4* in all post-mitotic neurons. In green, we see the intracellular domain carboxyl terminus of APPL and in red the secreted amino terminus of APPL. This brain is also stained with anti-Repo to mark glia (white). On the close-up panels (last row), we can clearly observe the colocalization between SAPPL and the glial marker, Repo. **(c and c')** These pictures are from a live snapshot of glial cells, around KC bodies, of 7-day-old fly brains expressing the DF probe only in glia. The left panel shows a control: *w*; UAS myr mCherry pH Luorin; repoGal4* fly; and the right panel shows the mutant: *Appldw*; UAS myr mCherry pH Luorin; repoGal4. n* = 6. **(d and e)** This graph and histogram represent the quantification of the volume of endolysosomal compartments in glia. They show that the endolysosomal compartments are smaller in glia of APPL-/- flies, *n* = 6, ****$p < 0.0001$, Mann–Whitney post hoc test. Underlying data can be found in the S1 Data file. APPL, APP Like; DF, double fluorescent; GFP, green fluorescent protein; KC, Kenyon cell; SAPPL, secreted portion of APPL.

## APPL regulates glial endolysosomal volume and debris clearance function

Considering the involvement of APPL in the regulation of the size of endosomes in neurons, we asked whether SAPPL may play a similar role in glia. We expressed the myr-DF probe specifically in glia and performed live imaging of 7-day-old control and *appl^d* brains. In contrast to neurons, the endolysosomal compartments of glia had a reduced volume compared to the controls (Fig 6C, 6C', and 6D), with no significant effects on their numbers (S7A Fig). The volume and number of degradative compartments analyzed from fixed data was not affected by the absence of APPL (S7B–S7E Fig). TEM analysis, however, revealed some glial disruptions. In control brains, most cortex glia seemed to be intact, and their extensions occupied the spaces between neuronal cell bodies (S7F Fig). In contrast, we occasionally observed irregular distribution of cortex glia between neuronal cell bodies, as well as cytoplasmic blebbing in these glia in APPL null brains, suggesting that glia were either unhealthy and/or dysfunctional in the absence of APPL (S7G Fig). These data suggest the exciting possibility that SAPPL may act as a neuronal signal to regulate endolysosomal trafficking in glia.

Studies on mouse brain lesion models showed increased levels of alpha-secretase A Disintegrin and Metalloprotease (ADAM)-17 and ADAM-10 in reactive astrocytes 7 days post-lesion [44]. In *Drosophila*, using a model of stabbing of the brain, Kato and colleagues showed that glia lose their ability to react to axonal lesions within 10 days after injury [45]. Therefore, taking into consideration these findings and our data showing a role of APPL during the first week of adulthood in the fly brain and its transfer from neurons to glia, we asked if APPL is required for glia to clear neuronal debris.

To investigate this, we labeled (olfactory receptor neurons) ORNs with GFP in control and *appl^d* flies and used the model of antennal ablation [46] (Fig 7A). After ablating both antennae of 5-day-old flies, we dissected their brain and imaged ORN axonal debris (GFP, green) in the antennal lobes of the adult fly brain. In control brains, axonal debris were almost completely cleared by 5 days after ablation. In contrast, loss of APPL caused a significant reduction in the clearance of the degenerative axons by glia in 5 days post-ablation (Fig 7B–7F) (S8 Fig). This defect was rescued by re-expressing, in an *appl* null background, either flAPPL or only SAPPL specifically in ORNs (Fig 8A–8C). To test the extent of the delay in clearance, we examined control and *appl* null brains at 8 days post-ablation and found that axonal debris still persist in *appl* mutants at this late stage (Fig 8D). Therefore, APPL is a neuronal signal required in glia to regulate their ability to clear neuronal debris.

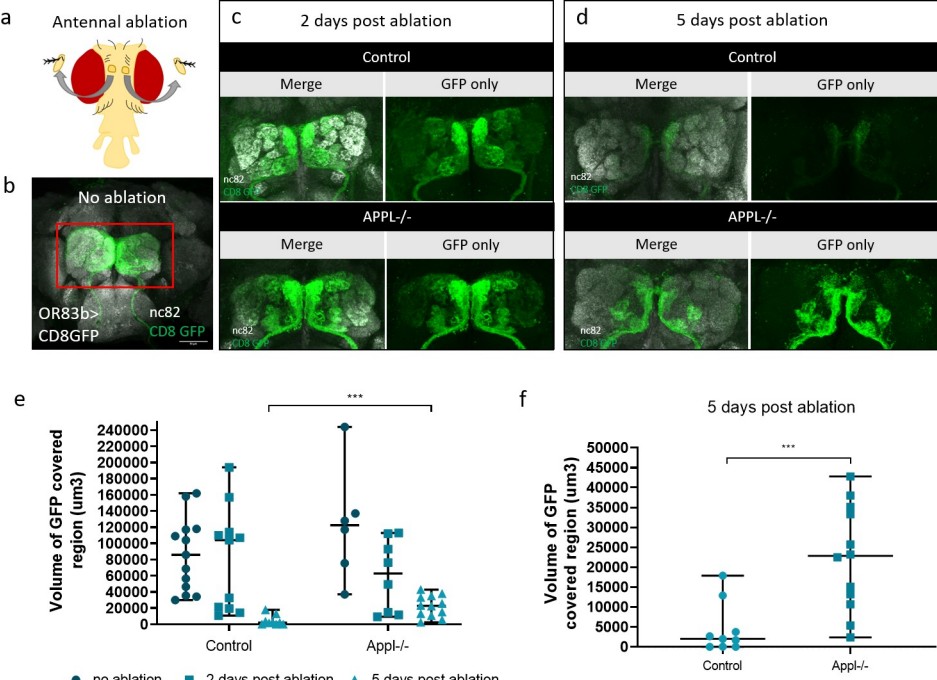

**Fig 7. APPL null flies show defective clearance of degenerating axons. (a)** Schematic presenting the head of a fly after antennal ablation. **(b–d)** Confocal images of GFP-labeled ORN axons at the antennal lobes of control: *w*/Y;* *OR83bGal4 UAS CD8 GFP/+;* and APPL-/-: *Appldw*/Y;OR83bGal4 UAS CD8 GFP/+;* flies before antennal ablation, 2 (c) and 5 (d) days after antennal ablation. The left panels of every section are also stained with nc82 to mark the neuropil. **(e)** Quantification of volume of GFP-covered region in the OR83b innervating glomeruli before, 2 and 5 days post-ablation, in control and APPL-/- flies. Every data point presented on the graph is the result of a brain. **(f)** We can observe that at 5 days post-ablation, the volume of GFP-covered region of axonal debris remaining in the APPL-/-brains is significantly higher comparing to the control, ***$p$ = 0.0009, Mann–Whitney post hoc test. $n$ = same as in (e). Underlying data can be found in the S1 Data file. APPL, APP Like; GFP, green fluorescent protein; ORN, olfactory receptor neuron.

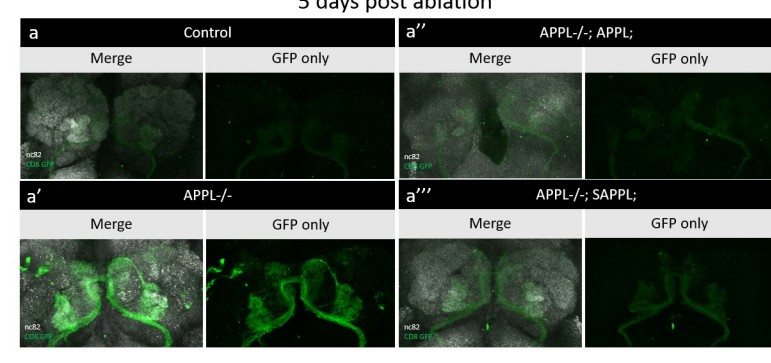

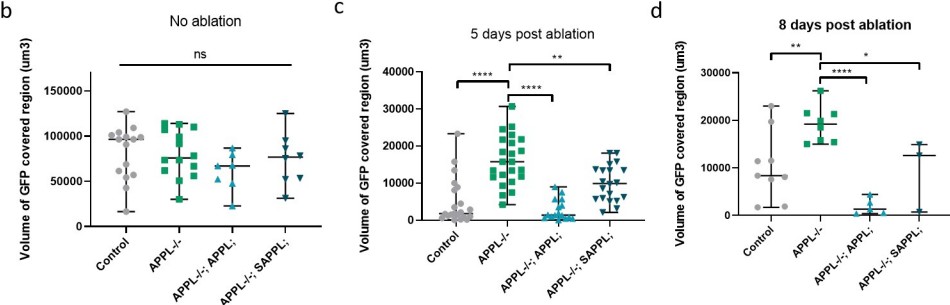

**Fig 8. Expression of flAPPL and SAPPL rescues glial clearance of axonal debris. (**a–a'''') Confocal images of GFP-labeled ORN axons at the antennal lobes at 5 days post-antennal ablation. These data show the rescue of a defective glial clearance of axonal debris, seen in APPL-/- flies, when we express the UAS APPL (a'') and UAS SAPPL (a''''). The left panels of every section are also stained with nc82 to mark the neuropil. (b–d) Quantification of volume of GFP-covered region (um3) in the OR83b innervating glomeruli before and at 5 (c) and 8 (d) days post-ablation, in control, APPL-/- and the rescue flies: *Appldw*; UAS APPL/OR83bGal4GFP;* and *Appldw*; UAS SAPPL/OR83bGal4GFP;.* (c) We can observe that at 5 days post-ablation, expressing APPL and SAPPL in an APPL-/- background are able to significantly rescue the defective glial clearance. One-way ANOVA, $F_{(3, 76)}$ = 23.13, df = 5, **$p$ = 0.0019, ****$p < 0.0001$. (d) This phenotype seems to have a similar pattern at 8 days post-ablation. One-way ANOVA, $F_{(3, 21)}$ = 10.47, df = 3, *$p$ = 0,04 **$p$ = 0.0099, ****$p < 0.0001$. For each genotype, every data point is an independent brain. Underlying data can be found in the S1 Data file. APPL, APP Like; flAPPL, full-length APPL; GFP, green fluorescent protein; ns, not significant; ORN, olfactory receptor neuron; SAPPL, secreted portion of APPL.

To investigate the mechanism of interaction between APPL and glia and the reason for defective glial clearance of axonal debris, we examined 2 distinct stages of reactive gliosis. Reactive gliosis is the brain's response to an injury or an infection. It starts with the induction of glial proliferation, migration to the site of injury, and finally, the activation of the glial Draper receptor for the engulfment of degenerating neurons or cellular debris [47].

We started by testing if APPL LOF affects glial migration to the site of the injury in control and *appl*[d] flies expressing GFP specifically in glia, before and 1 day after antennal ablation. We observed that loss of APPL did not have an impact on the migration of glia around and inside the antennal lobes (S9A–S9C Fig). Next, we examined the induction of the expression of the glial engulfment receptor, Draper, after injury. Although before antennal ablation the levels of Draper were similar in control and *appl* null brains, loss of APPL resulted in a significantly reduced activation of Draper 1 and 3 days after antennal ablation (Fig 9A–9E).

## Discussion

In this study, we took advantage of *D. melanogaster* to investigate and unravel the physiological function of APPL, the single fly homologue of the human APP, in the adult brain. Our key

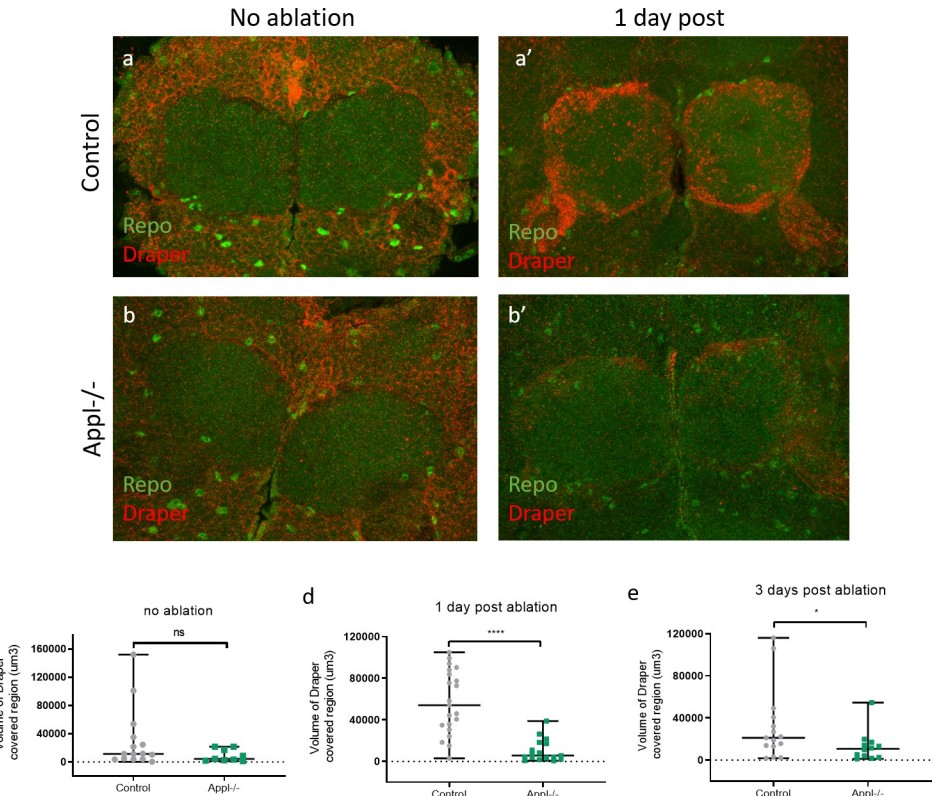

**Fig 9. Loss of APPL causes reduced Draper expression.** (a and a') Confocal images of the antennal lobes of approximately 7-day-old control fly brains stained with anti-Draper (red) and anti-Repo (green) at no ablation and 1 day after antennal ablation. (b and b') Confocal images of the antennal lobes of approximately 7-day-old Appl-/- fly brains at no ablation and 1 day after antennal ablation. (c) Quantification of volume of Draper-covered region (um3) around and inside the antennal lobes before and at 1 (d) and 3 (e) days post-ablation, in control and APPL-/- flies. We can observe that Draper expression/activation is reduced in the absence of APPL as there is significantly less volume of Draper-covered region around the antennal lobes at 1 day and 3 days post-ablation $^*p = 0.0427$, $^{****}p < 0.0001$, Mann–Whitney test. Underlying data can be found in the S1 Data file. APPL, APP Like; ns, not significant.

findings are (1) that APPL is required for neuronal survival during a critical period of early life; (2) that APPL regulates the size of endolysosomal vesicles in neurons and glia; and (3) that secreted APPL interacts with glial cells to enable the clearance of neuronal debris.

## APPL is required for adult brain homeostasis through the endolysosomal pathway

A homeostatic signaling system is composed of a set point, a feedback control, sensors, and an error signal. The error signal activates homeostatic effectors to drive compensatory alterations in the process being studied [48]. We propose a model (Fig 10) whereby the presence of APPL and its cleaved forms maintain the physiological flow of vesicular trafficking, either for degradation or for recycling, through the endolysosomal network in neurons. Simultaneously, in case of a system failure, a particular stress or an acute injury, there is an increased release of SAPPL, the error signal, activating degradation in glial cells, the homeostatic effector, to reset the system to its baseline.

It has been observed that *appl* null flies have a shorter life span and develop large neurodegenerative vacuoles in their brain by 30 days old [24]. In the present study, we demonstrate that the brain of *appl* null flies shows signs of dysfunctional homeostasis from a much younger

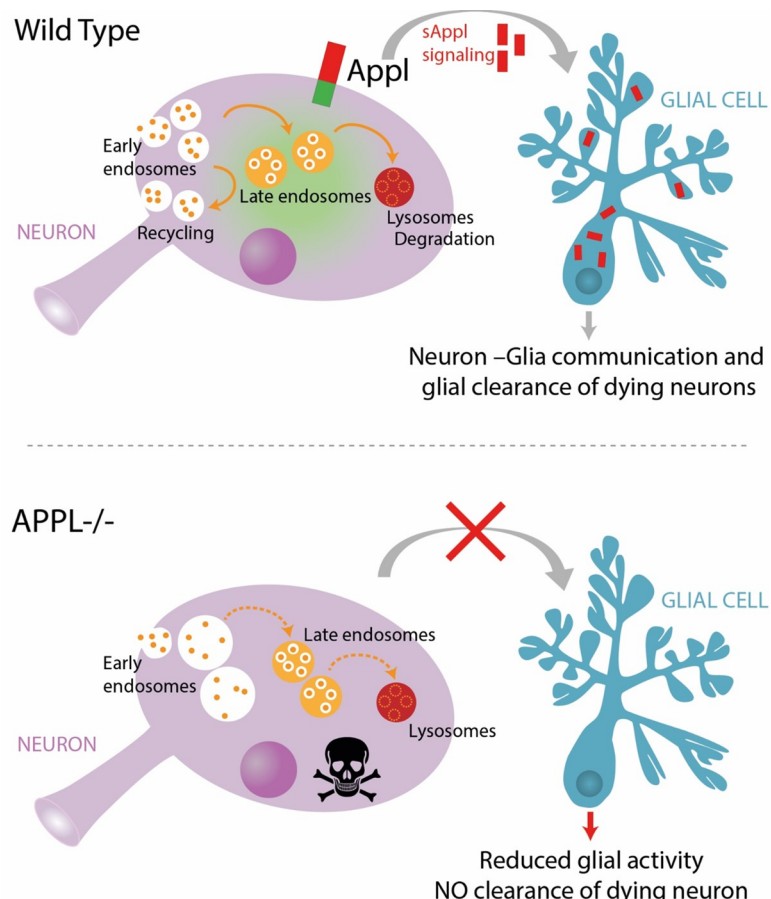

**Fig 10. Working model.** Under wild-type conditions, neuronal endolysosomal trafficking is maintained at normal equilibrium, in part by APPL function. Moreover, in case of neuronal stress or injury, the cleaved and secreted APPL, which continually communicates with glial cells, activates the glial clearance response. In contrast, under APPL LOF conditions, there is a dysregulation of the neuronal endolysosomal network, resulting in disrupted vesicular trafficking. Such a blockage causes the accumulation of protein cargo inside the cell, which is toxic and stressful for the neuron. However, as there is no SAPPL to communicate to glia that the neuron is dying, glial activity is reduced and degenerating neurons are not efficiently cleared. APPL, APP Like; LOF, loss of function; SAPPL, secreted portion of APPL.

age of 7 days old, resulting in a significantly increased number of apoptotic neurons and a significantly increased death rate from 20 days old.

Studies on Down syndrome (Trisomy 21), representing cases of elevated expression of APP, AD patient fibroblasts, AD mouse models, and recent studies using patients iPSCs have all shown evidence of a defective endolysosomal network [32,33,42]. In particular, neurons derived from AD patient iPSCs show that fAD mutations in APP or PSEN1 as well as knockout (KO) of APP, all cause alterations in the endolysosomal vesicle size and functionality. Some of the toxic effects on endolysosomal trafficking have been attributed not to amyloid accumulation but rather to the potential toxicity of the sAPPβ and/or APP β C-terminal fragment (APPβCTF), while a wealth of literature suggests that full-length APP (flAPP) and sAPPα are neuroprotective [24,49].

## APPL as a neuronal inducer of glial activity

Glial cells are the key immune responders of the brain that maintain neuronal homeostasis through neurotrophic mechanisms and by clearing degenerating neurons. Our data show that

neuronal expression of APPL is necessary and sufficient to activate glial clearance of neuronal debris and that glia take up neuronally released SAPPL. Moreover, we showed that the function of APPL in response to an injury involves regulating the glial engulfment receptor, Draper, via an as of yet unknown mechanism. It has also previously been shown that acute injury of the adult brain elicited an increased expression of APPL at and near the site of injury [50]. Interestingly, a recent study using iPSCs derived astrocytes with APP KO and fAD mutations revealed that loss of flAPP impairs cholesterol metabolism and the ability of astrocytes to clear Aβ protein aggregates [51]. Moreover, up-regulation of APP expression in neurons and α-secretase expression in reactive astrocytes was observed after the denervation of the mouse dentate gyrus [44]. Together, these observations indicate that the expression and proteolytic processing of APP are part of a neuro-glial signaling system responsible for monitoring brain health and activating glial responses to neuronal injury. Further future work will be needed to describe how exactly secreted APP fragments are taken up by glia and what cellular and molecular components they interact with and modify within glial cells to mediate appropriate levels of glial activation.

### Implications for neurodegeneration

Our findings that the complete loss of the *Drosophila* APP homologue causes deficits in the endolysosomal pathway, in neuron-induced glial clearance of debris and in neuronal death and organismal life span strongly suggest that, in the adult brain, the physiological function of flAPP and the consequences of fAD mutations are mechanistically related to one another. Furthermore, the fact that neuronal death and defective neuronal endosomes are observed very early in life of *appl* mutant flies further supports the notion that significant deficits exist in the AD brain long before any clinical symptoms appear. This may suggest that examining the size and/or function of the early endosome may identify risk for future neurodegeneration and offer future treatment pathways.

## Materials and methods

### Fly stocks and husbandry

**Fig 1.** Controls: Canton S (+/+;+/+;+/+), *yw*\*;;nsybGal4, w*;UAS CD8 GFP;, yw; + / +; + / +* kindly given by the lab of T. Preat. Appl-/-: *Appl^d w*;+/+;+/+* kindly given by the lab of J-M. Dura and *y1 sc* v1; P{TRiP.HMS01931}attP40; + / +* (UAS Appl RNAi with y+ as a marker) kindly given by the lab of T. Preat. Rescue experiment flies: *Appldw*; UAS APPL/+;* and *Appldw*; UAS SAPPL/+;*.

**Fig 2.** Control: *w*;UAS myr mCherry-pHLuorin;, yw*;;nsybGal4.* Appl-/-: *Appl^d w*; UAS myr mCherry pH Luorin; nsybGal4* kindly given by the lab of R. Hiesinger.

**Fig 3.** Controls: Canton S (+/+;+/+;+/+), *w*/Y;+/+;+/+.* Appl-/-: *Appl^d w*;+/+;+/+.*

**Fig 5.** Rab5-/+: *w*;Rab5 KO/CyO;* kindly given by the lab of R. Hiesinger.

**Fig 6.** Control: *w*;UAS myr mCherry-pHLuorin;, w*;;repoGal4* kindly given by V. Auld lab. Canton S (+/+;+/+;+/+). Appl-/-: *Appldw*;+/+;+/+.* Double fluorescent construct: *Appldw*hsflp / FM7C Df GmR YFP; UAS CherryApplGFP / CyO; (created in the lab), yw*;; nsybGal4.* Appl-/-: *Appl^d w*; UAS myr mCherry pH Luorin; repoGal4.*

**Fig 7.** Control:; *OR83bGal4 UAS CD8 GFP;* kindly given by the lab of I. Grunwald. Appl-/-: *Appldw*/Y;OR83bGal4 UAS CD8 GFP/+;.*

**Fig 8.** Rescue experiment flies: *Appldw*; UAS APPL/OR83bGal4GFP; and Appldw*; UAS SAPPL/OR83bGal4GFP;.*

- **S1 Fig.** *y1 sc* v1; P{TRiP.HMS01931}attP40; + / +, P{KK102543}VIE-260B, yw/Y; UAS CD8 GFP / +; nsybGal4 / +*

- **S2 Fig.** <u>Control</u>: *w\*/Y;+/+;+/+* <u>Appl-/-</u>: *w\* appl^td/Y;;*, and <u>Vang-/+</u>: *appl^td w\*/Y;Vang-/+;.*

- **S4 Fig.** <u>Rab7-/+</u>: *w\*;;Rab7 KO Crispr 3P3RFP/TM6B* kindly given by the lab of R. Hiesinger.

- **S6 Fig.** <u>dT expressed specifically in the retina</u>:; *UAS-mCherry-APPL-GFP/lexAop-CD4tdGFP; GMR-Gal4/Repo-lexA* kindly provided by the lab of R. Hiesinger.

All stocks were maintained using standard rich food at 21˚C, and all crosses and experiments were conducted at 25˚C on a 15h:9h light:dark cycle at constant humidity.

## Life span experiments

For the life span experiment, eclosing adults were collected under $CO_2$-induced anesthesia, over a 12-h period, and were left to mate for 48 h before sorting them into single sexes. After sorting, they were housed at a density of 15 flies per vial. Throughout the life span, flies were kept in a humidified, temperature-controlled, incubator with 15h:9h light:dark cycle at 25˚C on a standard, sucrose yeast corn and agar, media. Finally, they were transferred into new food and scored for death every 2 to 3 days throughout adult life [52].

## Immunochemistry

Adult brains were dissected in phosphate buffered saline (PBS) and fixed in 3.7% formaldehyde in PBT (PBS+Triton 0.3%) for 15 min. The samples were subsequently rinsed 4 times for 0', 5', 15', and 30' in PBT 0.3% and blocked in 1% bovine serum albumin (BSA) for at least 1 h. Following these steps, the brains were incubated with the primary antibody diluted in 1% BSA overnight at 4˚C. Then, the samples were rinsed 4 times for 0', 5', 15', and 30' in PBT 0.3% and were subsequently incubated with the appropriate fluorescent secondary antibodies in dark for 2 h at room temperature (RT). Finally, after 4 rinses with PBT 0.3%, the brains were placed in PBS and mounted on a polarized slide using Vectashield (Vector Laboratories, Burlingame, California, United States of America) as the mounting medium. Draper staining was performed as previously described [53].

The mounted fixed brains were imaged on an Olympus 1200 confocal microscope (Olympus France S.A.S., France) equipped with the following emission filters: 490 to 540 nm, 575 to 620 nm, and 665 to 755 nm.

The following antibodies were used: rabbit anti-cleaved Drosophila Dcp-1 (Cell Signaling Technology, Danvers, Massachusetts, USA, 1:100), rat anti-elav (Hybridoma bank, University of Iowa, Department of Biology, Iowa City, Iowa, USA, 1:100), mouse anti-repo (Hybridoma bank, 1:10), rabbit anti-repo (kindly given by Joachim Urban, 1:500), mouse anti-Draper8A1 (Hybridoma bank, 1:400), and mouse anti-nc82 (Hybridoma bank, 1:100).

## Transmission electron microscopy

First, we cut 7-day-old *Drosophila* adult heads and fixed them in 2% glutaraldehyde + 2% paraformaldehyde (PFA) + 1 mM CaCl2 in 0.1-M sodium cacodylate buffer (pH 7.4) for 1 h at RT. Following 3 rinses with sodium cacodylate buffer, we post-fixed samples with 1% osmium tetroxide in the same 0.1-M sodium cacodylate buffer for 1 h at RT. Then, we dehydrated them in a graded series of ethanol solutions (75%, 80%, 90%, and 100%, 10 min each). Final dehydration was performed twice in 100% acetone for 20 min. Subsequently, we infiltrated samples with Epon 812 (epoxy resin) in 2 steps: 1 night at +4˚C in a 1:1 mixture of Epon and acetone in an airtight container and 2 h at RT in pure Epon. Finally, we placed samples in molds with fresh resin and cured them in a dry oven at 60˚C for 48 h.

Blocs were cut in 1-μm semi-thin sections with an ultramicrotome EM UC7 (Leica, Buffalo Grove, Illinois, USA). Sections were stained with 1% toluidine in borax buffer 0.1 M. Then, we cut ultrathin sections (approximately 70-nm thick) and collected them on copper grid (Electron Microscopy Sciences, Hatfield, Pennsylvania). They were contrasted with Reynolds lead citrate for 7 min. Observations were made with a Hitachi HT 7700 electron microscope (Hitachi, Tokyo, Japan) operating at 70 kV. Electron micrographs were taken using an integrated AMT XR41-B camera (Brickfields Business Park, Suffolk, United Kingdom) (2048 × 2048 pixels).

### Adult brain culture and live imaging

Adult brains were dissected in cold Schneider's *Drosophila* Medium and mounted in the culture chambers perfused with culture medium and 0.4% dialyzed low-melting agarose [54]. Live imaging was performed at room temperature using a Leica TCS SP8 X confocal microscope (Leica) with a resonant scanner, using 63× water objective (+3.3 zoom). White laser excitation was set to 488 nm for pHLuorin and 587 nm for mCherry signal acquisitions [41].

### Quantification and statistical analysis

Imaging data were processed and presented using ImageJ (National Institutes of Health, Bethesda, Maryland, USA). ImageJ was also used for manual quantification of the apoptotic, dcp-1 positive cells slide by slide throughout the z-stack and for selecting regions of interest using the "ROI Manager" function. For the endolysosomal compartments analysis, we used the IMARIS software (Bitplane, Zurich, Switzerland), for both live and fixed images. To quantify the number and volume of the endolysosomal compartments, we used the Surface function, enabling the "Split touching objects" mode and keeping the same intensity threshold across samples and conditions. In the fixed images, to distinguish the red, acidic, compartments from the endosomes and quantify them, we used the "Spot colocalize" function. To measure the volume of the ones non-colocalizing, we used the Surface function enabling the "Split touching objects" mode. Finally, the IMARIS software (Bitplane) and, more specifically, the Surface function, was also used to quantify the volume of remaining GFP expressing axonal debris in the antennal ablation experiment, again using the same intensity threshold across samples and conditions (S9 Fig). Graphs were generated and statistical analysis was conducted using GraphPad Prism 8 (GraphPad Software, San Diego, California, USA).

### Olfactory receptor injury protocol

For the antennal ablation experiment, we used; *OR83b Gal4 UAS CD8 GFP;* flies, expressing GFP in most of the ORNs, and crossed them with control and *appl^d* background flies. The progeny of these crosses was collected daily and, after selecting the right genotype, we ablated both antennae of 5-day-old flies using finely sharpened tweezers. Then, we dissected the adult brains at 2, 5, and 8 days post-ablation and followed the immunostaining procedure, as previously described, in dark. We used anti-nc82 as the neuropil antibody in order to better visualize the antennal lobe glomeruli of the adult brain and focus our quantification of the endogenously expressed GFP-covered region accordingly.

## Supporting information

**S1 Fig. Levels of reduction of APPL expression correlates with the increase in Dcp1 positive cells in the brain of 7 days old flies.** (a) This graph shows the number of apoptotic cells in

the central brain of control,;; nsyb Gal4 and; UAS CD8 GFP; nsyb Gal4, and 2 different Appl RNAi flies at 7 days old. Each data point represents the number of apoptotic cells, Dcp-1 positive cells, in a single brain. For the analysis of these data, we used 1-way ANOVA with Tukey multiple comparison test $F(3,137) = 6.050$, df = 3, $^*p = 0.0176$, $^*p = 0.0252$, $^{**}p = 0.0012$. (b) This table describes the genotype of the 2 different RNAi lines used and the controls. (c–f') Confocal images of the central brain of each genotype stained with anti-App (green) to observe the expression levels of APPL. Underlying data can be found in the S1 Data file.
(TIF)

**S2 Fig. APPL does not seem to interact with the Wnt PCP pathway to maintain neuronal health.** Quantification of apoptotic cells in the central brain of control, $w^*/Y;+/+;+/+$, $w^* appl^d/Y;;$, and $appl^d w^*/Y;Vang-/+;$. Reducing 1 copy of Vang, a key member of the Wnt PCP pathway, in an APPL-/- background has no effect on the accumulation of apoptotic cells. Underlying data can be found in the S1 Data file.
(TIF)

**S3 Fig. Loss of APPL causes enlarged endolysosomal compartments in neurons.** (a) Histogram presenting the volume of each endolysosomal compartment in neurons of control, $w^*$; *UAS myr mCherry pH Luorin; nsyb Gal4* fly, and $appl^d$ mutant: *APPLd; UAS myr mCherry pH Luorin; nsybGal4*, flies. (b) This histogram presents the relative frequency of endolysosomal compartments in neurons. (c) This graph shows the quantification of the number of endolysosomal compartments/um3 in neurons, which is not significantly different between control and Appl-/- flies. (d) This graph shows that the number of degradative compartments/um3 is also not significantly affected by the absence of APPL, every dot corresponds to a brain. (e) These 4 panels represent the same data as in Fig 2D but separated in smaller groups of volume ranges; endolysosomal compartments of 0.004 to 0.35 um$^3$, 0.35 to 1 um$^3$, 1 to 2um$^3$, and 2 to 7 um$^3$. As observed, the most important differences in size are in the smallest volume group $^{****}p < 0.0001$ and the 1 from 1 to 2 um$^3$, $^{**}p = 0.003$, Mann–Whitney test. Underlying data can be found in the S1 Data file.
(TIF)

**S4 Fig. Increased autophagy in APPL null flies.** (a) Confocal stack of the central brain stained with Elav (red) to mark the neuronal cell bodies and Atg8 (green) to mark the autophagosomes. In this picture, we compare Canton S (control) flies to Appl-/-. (b) Graph presenting the increased number of Atg8 puncta per brain in appld background flies at 7 of age. Mann–Whitney test $^{****}p < 0.0001$. (c) This graph represents the quantification of apoptotic cells in the brain of appld background flies with 1 reduced copy of the Atg7 gene. (d) Graph presenting the quantification of Dcp-1 positive cells in the brain of 7-day-old APPL-/- flies lacking 1 copy of Rab7, the late endosome marker, $w^* appl^d/Y;;Rab7\ KO\ Crispr\ 3P3RFP/+$, compared to control and Appl-/- flies. This graph shows no difference in the number of apoptotic neuronal cell death when 1 copy of Rab7 is reduced. (e) Life span analysis of control and $appl^d$ flies lacking 1 copy of Rab7. This survival curve reveals that reducing 1 copy of Rab7 in an $appl^d$ background increases significantly the death rate of Appl-/- flies, starting from an even earlier age and reducing the overall life span of $appl^d$ flies, 2-way ANOVA with Tukey multiple comparison test $^{**}p = 0.0022$, $^{****}p < 0.0001$. Underlying data can be found in the S1 Data file.
(TIF)

**S5 Fig. APPL regulates the size of early endosomes in neurons.** (a) Histogram presenting the volume of each early endosome–like vesicle in neurons of control, $+/+;+/+;+/+$ and APPL-/- flies, $w^* appl^d/Y;;$ flies. (b) This histogram represents the relative frequency of early endosome–like vesicles and their size in APPL-/- flies, $w^* appl^d/Y;;$, comparing to control, $+/+;$

+/+;+/+. c) This histogram represents the relative frequency of lysosomes and their size in APPL-/- flies, $w^*appl^d/Y$;;, comparing to control, +/+;+/+;+/+. (d) Histogram representing the relative frequency of lysosomes per cell slice in APPL-/- flies, $w^*appl^d/Y$;;, comparing to control, +/+;+/+;+/+ fly brains. (e) Focusing on the 7-day-old time point, which showed significantly enlarged early endosome–like vacuoles in Appl-/- flies (Fig 3D), we now knock down the expression of APPL only in neurons using the *yw; UAS APPL RNAi (y+); nsybGal4* and find a similar increase in the size of these early endosome–like vacuoles comparing to the control: *yw;;Gal4nsyb/+*. $n$ = 3 to 5 brains per genotype and approximately 60 cells analyzed per genotype, ****$p$ < 0.0001. (f) As in (Fig 3E), the percentage of cells with endosomes was significantly higher in *yw; UAS APPL RNAi (y+); nsybGal4* comparing to control. $n$ = 3 to 5 brains per genotype and approximately 60 cells analyzed per genotype, Fisher exact test ****$p$ < 0.0001. (g) TEM horizontal sections of the cortical region of 7-day-old fly brains showing neuronal cell bodies (circled in blue) and the early endosome–like vacuoles marked with a red arrow. $n$ = 3 to 5 brains per genotype and approximately 60 cells analyzed per genotype. Underlying data can be found in the S1 Data file.
(TIF)

**S6 Fig. SAPPL travels ubiquitously regardless the site of expression of APPL.** (a) Confocal sections of a control fly brain throughout development until adulthood that expresses the double fluorescent tagged APPL construct specifically in the retina using the GMR Gal4 driver,*; UAS-mCherry-APPL-GFP/lexAop-CD4tdGFP; GMR-Gal4/Repo-lexA*. As we can see between P50 and P80, there is a significant release of SAPPL (white) beyond the site of expression reaching all areas of the brain. (b) This is a different experiment using these flies:*; UAS-mCherry-APPL-GFP; GMR-Gal4*. These close-ups on the photoreceptors confirm that it is only the SAPPL (white) that travels ubiquitously in the brain, although the carboxyl terminus of APPL (green), the intracellular part, remains in the cell bodies where it is being expressed. (c) This graph shows the adult stage of the flies used in (a) and highlights that the SAPPL (white) not only travels throughout the brain but also colocalizes specifically with the glial marker, Repo (green). (d and e) Rescue experiment of the loss of mushroom body (MB) β lobes (red oval) phenotype in *Appld;;nsybGal4* flies with a 12.2% phenotype penetrance ($n$ = 49). *Appld; UAS mCherry Appl GFP; nsybGal4* is functional as it rescues this loss of MB β lobes with a 0% phenotype penetrance ($n$ = 39). ***$p$ = 0.0002 Fisher exact test. Underlying data can be found in the S1 Data file.
(TIF)

**S7 Fig. APPL effects on glial endolysosomal network.** (a, c) The number of endolysosomal and degradative/acidic compartments were not affected by the absence of APPL in glia. (b and b') These confocal slices represent the same area of glial cells but this time from a fixed tissue of control and APPL-/- 7-day-old fly brains. (d and e) The volume of degradative/acidic compartments was also similar between both conditions. (f and g) TEM horizontal sections of the cortical region of a 7-day-old fly brain showing neuronal cell bodies and cortical glia (circled in blue) between them. We can observe that the distribution of cortical glia in the brain of APPL-/- flies is abnormal, they have an irregular shape and many vesicles, comparing to the control. Underlying data can be found in the S1 Data file.
(TIF)

**S8 Fig. Glial clearance of axonal debris at 2 days post-ablation.** Quantification of volume of GFP-covered region (um3) in the OR83b innervating glomeruli at 2 days post-ablation, in control, APPL-/- and the rescue flies: *Appldw*; UAS APPL/OR83bGal4GFP*; and *Appldw*; UAS SAPPL/OR83bGal4GFP*;. Underlying data can be found in the S1 Data file.
(TIF)

**S9 Fig. Loss of APPL does not affect glial migration to the site of injury. (**a and a') Confocal images of a control brain; *UAS GFP/+;Repo Gal4/+* before and 1 day after ablation. The endogenous expression of GFP represents the glial migration around and inside the antennal lobes after antennal ablation. (b and b') In Appl null flies *Appld;UAS GFP/+;Repo Gal4/+*, we observe a similar reaction of glial cells after antennal ablation. (c) This graph presents the quantification of the volume (um3) of GFP-covered area in and around the antennal lobes before and 1 day after ablation in control compared to Appl-/- flies. Underlying data can be found in the S1 Data file.
(TIF)

**S1 Data. Excel file with all quantification data corresponding to each graph on every figure.**
(XLSX)

**S1 Movie.**
(M4V)

## Acknowledgments

We thank the Bloomington stock center (NIH P40OD018537) for providing flies used in this study. We thank all members of the Hassan and Hiesinger labs for support and valuable comments. More specifically, we would like to thank Corentine Marie for her help in the illustration of our working model. We are grateful to the ICM imaging facility ICM.Quant for imaging and image analysis support.

## Author Contributions

**Conceptualization:** Irini A. Kessissoglou, Bassem A. Hassan.

**Formal analysis:** Irini A. Kessissoglou, Dominique Langui, Amr Hasan, Maral Maral.

**Funding acquisition:** Peter Robin Hiesinger, Bassem A. Hassan.

**Investigation:** Irini A. Kessissoglou, Peter Robin Hiesinger, Bassem A. Hassan.

**Methodology:** Irini A. Kessissoglou, Dominique Langui, Amr Hasan, Maral Maral, Suchetana B. Dutta.

**Supervision:** Peter Robin Hiesinger, Bassem A. Hassan.

**Writing – original draft:** Irini A. Kessissoglou, Bassem A. Hassan.

**Writing – review & editing:** Irini A. Kessissoglou, Amr Hasan, Peter Robin Hiesinger, Bassem A. Hassan.

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
