## [Editor Report · Decision Letter 0]

26 Feb 2020

Dear Bassem, 

Thank you for submitting your manuscript entitled "The Drosophila Amyloid Precursor Protein homologue mediates neuronal survival and neuro-glial interactions" for consideration as a Research Article by PLOS Biology.

Your manuscript has now been evaluated by the PLOS Biology editorial staff, as well as by an Academic Editor with relevant expertise, and I am writing to let you know that we would like to send your submission out for external peer review.

Please re-submit your manuscript within two working days, i.e. by Feb 28 2020 11:59PM.

Kind regards,

Gabriel Gasque, Ph.D.,

Senior Editor

PLOS Biology

---

## [Decision Letter · Decision Letter 1]

2 Apr 2020

Dear Bassem,

Thank you very much for submitting your manuscript "The Drosophila Amyloid Precursor Protein homologue mediates neuronal survival and neuro-glial interactions" for consideration as a Research Article at PLOS Biology. Your manuscript has been evaluated by the PLOS Biology editors, by an Academic Editor with relevant expertise, and by three independent reviewers.

In light of the reviews (below), we will not be able to accept the current version of the manuscript, but we would welcome re-submission of a much-revised version that takes into account the reviewers' comments. We cannot make any decision about publication until we have seen the revised manuscript and your response to the reviewers' comments. Your revised manuscript is also likely to be sent for further evaluation by the reviewers.

We expect to receive your revised manuscript within 2 months. 

**IMPORTANT - SUBMITTING YOUR REVISION**

Your revisions should address the specific points made by each reviewer. As you will see, the reviewers think your findings are potentially interesting, but also express some concerns. They have also provided clear guidance on how these should be addressed, and which we think you should embrace. Having discussed these comments with the Academic Editor, we have the following directions, which are pointers, but no exhaustive:

1) Two reviewers seem to be concerned about the effect sizes, which are small, particularly on cell death and on endolysosomal volumes. We think that reviewer 3’s suggestions are quite appropriate. Please do the additional controls, such as rescue of the glial disruptions and endolysosomal phenotypes. This, we think, could eliminate the major concern that subtle differences in genetic background are at play here. In general, reviewer 3’s comments are sharp, we think, and you should address them thoroughly, including increasing the sample size in some experiments. 

2) You should show higher magnification images for figure 3, as you have done elsewhere.

3) Reviewer 1 also raises a series of critiques that we think have more to do with interpretation of the data. We think that these sorts of critiques could be addressed with a more nuanced discussion of the data.

4) The Academic Editor also thinks, without wanting to be too instructive, that you could use the CPV reporter to mark apoptotic cells and glial cells. This is a UAS-driven construct that has reporters that are activated by caspase cleavage. I am passing this recommendation to you, in case you find it useful. 

Please submit the following files along with your revised manuscript:

*Re-submission Checklist*

*Published Peer Review*

*PLOS Data Policy*

*Blot and Gel Data Policy*

Sincerely,

Gabriel Gasque, Ph.D., 

Senior Editor

PLOS Biology

REVIEWS:

Reviewer #1: The manuscript by Kessissoglou et al. aims at finding a physiological function of APPL in the adult Drosophila brain. The authors analyze appl null mutant and present here different aspects of its phenotype, such as lifespan, cell death in the adult brain, dimensions of endosomal and degradative compartments in neurons and glia and evaluation of glial capacity to clear injured axons. The described results are interesting but too preliminary for making the presented conclusions. 

The authors show that appl mutant flies exhibit a significantly reduced lifespan and accumulation of Dcp-1-positive cells in the adult brain as compared to control flies. However, the presented figures of the anti-Dcp-1 staining are not convincing and the differences between control and mutant flies are based on an extremely low number of apoptotic cells in the adult brain (2-8). In addition, this staining often shows a strong background like in Fig. 3i, which therefore cannot be used as the solitary detection and quantification of dying cells. Moreover, accumulation of apoptotic cells often results from their impaired clearance and not necessarily from the excessive cell death. It is not clear from the presented data whether excessive neuronal death actually occurs in the appl mutant.

Based on the assessment of the volume of endosomal and degradative compartments in neurons, the authors claim their enlargement in the appl mutant brains compared to control. However, the live imaging pictures are not of sufficient quality to quantify fluorescence and the number of analyzed brains is very low (2). It is not clear to me whether and how the slightly increased volume of endosomal compartments leads to neuronal cell death.

The reduced capacity of glia to clear degenerating axons in the appl mutant, which is rescued by the full length APPL or SAPPL is an interesting result. However, it does not certainly indicate the impaired glial ability to clear apoptotic neurons in the appl mutant. These are two distinct processes, which share some common mechanisms. 

Reviewer #2: Kessissoglou et al. provide compelling evidence for a role of the fly homologue of APP in brain homeostasis and neuron-glia cross-talk. They show that loss of APPL results in reduced lifespan, increased neuronal cell death, enlarged early-endosomal size in neurons, and reduced endolysomal size in glia. By expressing a double-tagged APPL construct in neurons of APPL -/- flies, they determine that secreted APPL is taken up by glia. Finally, expression of sAPPL in neurons of APPL -/- flies rescues glial-mediated clearance of axonal debri in an antennal ablation model. These findings are of high interest for its contribution to understanding APP biology and its potential implications for Alzheimer's disease. The role of sAPP in neuron-glia cross-talk is highly novel and will be of particular interest to the field. Therefore, I find this study by Kessissoglou et al to be appropriate for publication in PLOS Biology, if the following minor points can be addressed:

1) Statistics are needed in Figure S4. The difference in the survival curves between control and Appl is striking in Figure 1A. However, this difference does not seem to be replicated in Figure S4. What is the explanation for this? Overall, it is difficult to conclude from these data if there is rescue of lifespan with loss of Rab5 because the effect of APP -/- is not obvious.

2) In Figure 2D, the n should be increased to at least 3 brains/genotype. In Figure 2 F-G and 4D-E, it is unclear how many brains/genotype were analyzed.

3) Including a plot of the relative frequency for the data shown in 2D would be helpful to interpret the differences between the genotypes (similar to what was done for data in 2F and 4E). 

4) It is unclear why there would be volumes = 0 um plotted in Figures 2D,2F,2G and 4D,E. Were all cells counted and those with no compartments were quantified as 0? If so, perhaps a better way to plot it would be to plot the % of cells with endolyosomal compartments, and then separately plot the volume for those >0. Alternatively, a size cut-off for what is considered an endolyosomal compartment may be needed. 

5) In Figure 3i, It seems the background of the Dcp-1 staining is higher in control than Appl -/-. Could the effect observed be an artifact? Do you have representative images with similar background staining in which this difference in Dcp-1 positive cells is still observed? 

6) The manuscript is very well-written. Only a few, very minor suggested edits to the text:

 a) In Figure legend 1, should ***p=0.0078 be instead **p=0.0078?

 b) In Figure 3H, the value for *p should be added.

 c) A scale bar is needed in Figure 2E. 

 d) The order of figures S5 and S6 should be switched, since S6 comes before S5 in the text.

 e) There is no reference to Figures S2a-b in the text. 

 f) In the text (lines 180-191) it reads that in both Figure 4a and Figure S6 APPL was expressed in the retina; however, from the figure legends it seems that while APPL was expressed in the retina in Figure 4 , APPL was expressed in the optic lobe in Figure S6 using a different Gal4 driver- please clarify appropriately in the text. 

 g) I'd suggest changing "strange shape" to "abnormal" or "irregular" shape in Figure 4 Legend. 

Reviewer #3: The manuscript represents an important contribution to molecular understanding of the fly APP homologue, Appl. Because the fly Appl has homology to human APP, these findings may be relevant to Alzheimer's disease. Overall the manuscript is well written, and combines data from genetic manipulations with immunohistochemistry and transmission electron microscopy. To test the potential implication of Drosophila Appl in the endolysosomal pathway and study its functions as signaling molecule, the authors used a loss of function Drosophila mutant in the gene. They found a small increase in cell death in the brain of 7d animals. They examined the endolysosomal organelles and found that these were larger in the appl mutant neurons. They also report interactions with rab5 and rab7. By tagging a transgene of appl with GFP on the inside and mCherry on the outside, they show that the mCherry appears in glia. They then functionally show that the ability of axonal fragments generated upon neural body ablation (removal of the antennal lobes) is delayed in appl mutants. Adding Appl back to neurons rescues this effect. The authors suggest a model where Appl and its cleaved forms maintain the physiological flow of vesicular trafficking through the endolysosomal network in neurons and acts as signaling molecule to enable glial cells to remove the neuronal debris in the case of axotomy.

Overall these are interesting data that will be of interest to the field. They suggest an interesting role of appl, and an interplay between neurons and glia in Wallerian degeneration that will be of interest. 

A number of issues, however, need to be addressed to make the results solid and compelling: 

The biggest concern regarding these data is that the effect is rather subtle and so the controls must be very rigorous to rule out that the effects they are seeing are simply due to genetic background. A number of comments address this. 

The authors show that the effect on cell death number in the brain is replicated by an Appl RNAi line. The line needs to be validated that it reduces Appl, and all of the other assays should also be completed with this independent approach to reducing Appl for rigor. Alternatively, they could use multiple alleles of Appl. 

Related to this, in Fig.1c the average of apoptotic cells (DCP1 positive cells) in 7d Control was 2 and in Appl -/- flies of the same age was 7. Whereas on Fig. S1 the average of DCP1 positive cells in 7d Control was 5 and in Appl -/- flies of the same age was still 7. Considering the discrepancies, it is not clear how robust the difference between Control and mutant is, and if it is enough to reduce the lifespan and induce neurodegeneration. 

The authors demonstrate strong glial disruptions and reduced volume of endolysosomal compartments in appl null flies. The authors should re-express APPL and SAPPL and rescue these phenotypes in the appl null background, to show that the effects are due to appl and not genetic background. 

For the quantification of the volume of endolysosomal compartments (Fig.2d) only 2 brains/genotype were used for analysis. It is standard to use at least 3 biological specimens and to reproduce all experiments three times or more independently. Along the same lines, it is not clear how many biological replicates were used in experiments listed on Fig.3c, Fig. 4d, Fig. 5e, f, Fig.6b,c,d. 

The rab5 and rab7 interactions should be shown with 2 different alleles as well, to rule out background effects. 

Does dT-APPL rescue the mutant, is this form of the protein functional? This should be addressed. 

What happens if they just express the APPL in the glia in the axotomy model? Potentially this may rescue and would confirm a role in glia. 

Other comments: 

Sometimes the authors mention protein forms without explaining them (sAPPL). 

They use SAPPL, sAPPL and APPLS. Which is it? Are those all refer to the same form of the protein?

---

## [Decision Letter · Decision Letter 2]

25 Sep 2020

Dear Bassem,

Thank you for submitting your revised Research Article entitled "The Drosophila Amyloid Precursor Protein homologue mediates neuronal survival and neuro-glial interactions" for publication in PLOS Biology. I have now obtained advice from the original reviewers and have discussed their comments with the Academic Editor. 

Based on the reviews, we will probably accept this manuscript for publication, assuming that you will modify the manuscript to address the remaining points raised by reviewers 1 and 3, with additional quantifications and textual changes. Please also make sure to address the data and other policy-related requests noted at the end of this email.

We expect to receive your revised manuscript within two weeks. Your revisions should address the specific points made by each reviewer. In addition to the remaining revisions and before we will be able to formally accept your manuscript and consider it "in press", we also need to ensure that your article conforms to our guidelines. A member of our team will be in touch shortly with a set of requests. As we can't proceed until these requirements are met, your swift response will help prevent delays to publication.

- a cover letter that should detail your responses to any editorial requests, if applicable

*Copyediting*

*Published Peer Review History*

*Early Version*

Sincerely,

Gabriel Gasque, Ph.D.,

Senior Editor,

ggasque@plos.org,

PLOS Biology

DATA POLICY:

Note that we do not require all raw data. Rather, we ask for all individual quantitative observations that underlie the data summarized in the figures and results of your paper. For an example see here: http://www.plosbiology.org/article/info%3Adoi%2F10.1371%2Fjournal.pbio.1001908#s5

These data can be made available in one of the following forms:

Regardless of the method selected, please ensure that you provide the individual numerical values that underlie the summary data displayed in the following figure panels: Figures 1acdef, 2dfg, 3c-h, 4bc, 5bc, 6de, 7ef, 8b-d, 9c-e, S1a, S2, S3a-e, S4b-e, S5a-f, S6d, S7acde, S8, and S10c.

Please also ensure that each figure legend in your manuscript includes information on where the underlying data can be found and that your supplemental data file/s has/have a legend.

Reviewer remarks:

Reviewer #1: The authors addressed most of my concerns. I have two comments:

- Figure legend must be added to Figure 10

- On the bottom panel of Figure 10 the authors suggest that overexpression of APPLoe leads to glial overactivation and clearance of healthy neurons, which is not consistent with their results described in line 111: "…In contrast, overexpressing SAPPL in a control background caused the presence of significantly more apoptotic cells in the brain of 7 days old flies (Figure 1f).", which means that it leads to more dying neurons and not to clearance of healthy ones. It should be changed accordingly.

Reviewer #2: The authors have appropriately addressed my critiques, and I find this manuscript suitable for publication in PLOS Biology. 

Reviewer #3: The authors have performed considerable revisions that have increased the rigor of their data, and done a lot to make the data far more compelling. It is an interesting story, and an important finding. The interaction with draper, to increase its protein expression and trigger phagocytosis is interesting.

There are a number of minor issues to be addressed/resolved.

line 21: in the abstract they say that loss of appl causes dysregulation of endolysosomal function, in both neurons and glia. This is somewhat confusing, as they well document the defect in neurons, but the impact on glia they argue must be non-autonomous. Glia could be removed from this sentence for clarity.

line187: They refer to rab5 immunostaining as "significantly broader". It is not clear what that means - in more cells than it normally is? or do they mean the level is higher? this should be more clear and quantitated. 

line 218: in figure S6e, could they point out the MB loss. 

The addition of TEM data is nice, but in some cases it needs quantitation and they need to clarify how many animals they looked at. 

For example, in figure 6g, how many times was this seen? To rule out an artifact like bad fixation. 

In figure 10, they add a final panel bottom whereby they indicate that upregulation of APPL induces over activation of glia and clearance of healthy neurons. This part of the model is likely incorrect, given they are driving expression of APPL in a rescue setting by the GAL4-UAS system at levels that are undoubtedly much higher than normal levels and yet they see only rescue. This should probably be removed or a legend should be added that this is speculative. Or they could do an experiment, with the ORNs showing active removal of neurons when up regulating APPL and upregulation of draper on all the glia. The data they have on an up regulation effect is Figure 1f, where they suggest that up regulation of sAPPL causes more cell death in the brain (mimicking the loss of function).

---

## [Editor Report · Decision Letter 3]

2 Nov 2020

Dear Dr Hassan,

On behalf of my colleagues and the Academic Editor, Josh Dubnau, I am pleased to inform you that we will be delighted to publish your Research Article in PLOS Biology. 

PRODUCTION PROCESS

Before publication you will see the copyedited word document (within 5 business days) and a PDF proof shortly after that. The copyeditor will be in touch shortly before sending you the copyedited Word document. We will make some revisions at copyediting stage to conform to our general style, and for clarification. When you receive this version you should check and revise it very carefully, including figures, tables, references, and supporting information, because corrections at the next stage (proofs) will be strictly limited to (1) errors in author names or affiliations, (2) errors of scientific fact that would cause misunderstandings to readers, and (3) printer's (introduced) errors. Please return the copyedited file within 2 business days in order to ensure timely delivery of the PDF proof. 

If you are likely to be away when either this document or the proof is sent, please ensure we have contact information of a second person, as we will need you to respond quickly at each point. Given the disruptions resulting from the ongoing COVID-19 pandemic, there may be delays in the production process. We apologise in advance for any inconvenience caused and will do our best to minimize impact as far as possible.

EARLY VERSION

PRESS 

Kind regards,

Alice Musson

Publishing Editor, 

PLOS Biology

on behalf of

Gabriel Gasque,

Senior Editor

PLOS Biology